# Characterization of Winter Dysentery Bovine Coronavirus Isolated from Cattle in Israel

**DOI:** 10.3390/v13061070

**Published:** 2021-06-04

**Authors:** Dan David, Nick Storm, Waksman Ilan, Asaf Sol

**Affiliations:** 1Kimron Veterinary Institute, Bet Dagan 50250, Israel; Dand@moag.gov.il (D.D.); StanislavI@moag.gov.il (N.S.); 2Hachaklait, Veterinary Services, Caesarea 3079548, Israel; waksman@hachaklait.co.il

**Keywords:** bovine coronavirus, winter dysentery, spike, S1, hemagglutinin esterase

## Abstract

Bovine coronavirus (BCoV) is the causative agent of winter dysentery (WD). In adult dairy cattle, WD is characterized by hemorrhagic diarrhea and a reduction in milk production. Therefore, WD leads to significant economic losses in dairy farms. In this study, we aimed to isolate and characterize local BCoV strains. BCoV positive samples, collected during 2017–2021, were used to amplify and sequence the S1 domain of S glycoprotein and the full hemagglutinin esterase gene. Based on our molecular analysis, local strains belong to different genetic variants circulating in dairy farms in Israel. Phylogenetic analysis revealed that all local strains clustered together and in proximity to other BCoV circulating in the area. Additionally, we found that local strains are genetically distant from the reference enteric strain Mebus. To our knowledge, this is the first report providing molecular data on BCoV circulating in Israel.

## 1. Introduction

Winter dysentery (WD) is a sporadic, acute, contagious hemorrhagic enterocolitis of adult cattle. WD is characterized by epizootic outbreaks of self-limiting hemorrhagic diarrhea, high fever, depression, dehydration, anorexia, colic and decrease in milk production [1,2,3,4,5]. WD outbreaks are predominantly seen in postpartum adult dairy cows, with high morbidity rates. Diseased cows usually experience a marked drop in milk production, which leads to significant economic losses [6,7].

The causative agent of WD is bovine coronavirus (BCoV), first discovered in 1972 [8]. BCoV belongs to the order Nidovirales, family Coronaviruses and subfamily Coronavirinae within the genus *Betacoronavirus*. BCoV is an enveloped virus with a single-stranded non-segmented positive-sense RNA genome with five structural proteins: (S) spike glycoprotein, (M) membrane protein, (HE) hemagglutinin esterase glycoprotein, (E) small membrane protein and (N) nucleocapsid phosphoprotein [9]. The S glycoprotein is composed of two subunits: S1 and S2. Upon viral uptake, S1 binds the receptor on target cells while the S2 domain facilitates the fusion of viral and host membranes, which allow the viral genome to enter the host cell [10]. BCoV infection is initiated by HE and S protein attachment to target tissues and both are important for BCoV pathogenesis [11,12,13]. The hypervariable region of the S protein is used to determine the genetic variability and evolution of BCoV viruses, and S protein sequence alignments are important in evaluating novel variants of BCoV strains [14,15,16,17,18,19].

In Israel, there are approximately 100,000 dairy cattle. Although the presence of a viral agent causing WD clinical signs was first suggested in 1959 in a report by Komarov et al. [20], limited information is available about BCoV circulating in Israeli dairy farms. In this study, we isolated local strains of BCoV circulating in local farms. We cultured these strains, measured their replication in vitro and characterized them by molecular analysis of the S and HE proteins. To our knowledge, this is the first study reporting HE and partial S nucleotide sequences of BCoV from Israel.

## 2. Materials and Methods

### 2.1. Clinical Samples

During 2017–2021, a total of 853 fecal and rectal swabs were obtained from diseased cattle aged between 2 and 6 years. These samples were collected from large dairy farms scattered in Israel. Samples were collected from animals with distinct BCoV clinical signs, including hemorrhagic diarrhea and a significant decrease in milk production. Samples were further processed as follows: feces samples (~1 g) were diluted 1:10 in phosphate-buffered saline (PBS) and rectal swabs were suspended in 2 mL PBS. All samples were further incubated for 20 min at room temperature and then centrifuged at 1000 rpm for 10 min. Following centrifugation, supernatant from each sample was kept at −80 °C for future analysis. 

### 2.2. RNA Extraction and BCoV Molecular Detection

Supernatants of each sample were tested for the presence of BCoV RNA by targeting the BCoV nucleocapsid gene using primers and probes published by Kishimutu et al., 2016 [21]. Briefly, 100 µL of each sample was used to extract RNA using the MagMAX CORE Nucleic Acid Purification Kit (Applied biosystems, Carlsbad, CA, USA) and KingFisher Duo Prime (Thermo scientific, Carlsbad, CA, USA), according to the manufacturer’s instructions. Purified RNA was used to perform qRT-PCR using the Quanta qScript XLT 1-Step RT-qPCR ToughMix KIT (Qunatabio, Beverly, MA, USA) and analyzed using the Bio Rad CFX 96 Real Time detection system (Bio Rad, Hercules, CA, USA).

### 2.3. Virus Isolation

Following qRT-PCR, selected positive samples were filtered by 0.45 µm membrane filter and used to isolate BCoV by serial passaging of samples with human rectal tumor—HRT-18G cell line (ATCC^®^ CRL11663™), seeded in 25 cm^2^ flasks (Nunc, USA). Briefly, 1 mL from the post-filtered sample was applied on HRT-18G monolayer and incubated for 1 h at 37 °C and 5% CO_2_. Following incubation, cells were kept on maintenance medium containing Dulbecco’s modified Eagle’s medium (DMEM) (ATCC^®^ 30-2002™) supplemented with 2% fetal bovine serum (Gibco, Invitrogen, CA, USA). Cells were inspected daily for the presence of a cytopathic effect (CPE).

### 2.4. Replication of BCoV in HRT-18G

For the replication assay, we infected the HRT-18G cell line with BCoV isolate ISR2182 (Table 1). Briefly, the source feces samples were filtered as described above and diluted 1:10 in PBS. Then, 1 mL of diluted sample was used for infection. Following infection, we collected cells at different time points, extracted RNA and performed qRT-PCR as described above.

### 2.5. Immunofluorescence

HRT-18G cells were grown to confluence in Lab-Tek 8-well coverslip-bottom chamber slides (Nunc, San Diego, CA, USA). The wells were washed with PBS and then infected with BCoV isolate ISR2182, which was grown for 13 passages in HRT-18G and diluted 1:100 in PBS. Then, 100 µL from the diluted sample was incubated with HRT-18G for 45 min and maintenance medium was added as described above. Infection was carried out for 48 h. Following infection, cells were washed three times with PBS and fixed by cold acetone for 20 min at −20 °C. Cells were washed three times with PBS to aspirate the fixative and then incubated overnight at 4 °C with 1:100 monoclonal anti-nucleocapsid of BCoV (BC 6-4-A, RTI, Brookings, SD, USA) and kept in the dark. Next, primary antibody was removed by three washes in PBS and then 1:100 FITC-conjugated secondary antibody (Jackson immune-research Laboratories, West Grove, PA, USA) was added for 45 min in the dark. Nuclei were stained with DAPI (Sigma, Jerusalem, Israel) and slides were analyzed using Nikon fluorescence microscopy and NES elements software (Nikon Eclipse TE200-U).

### 2.6. PCR and Sequencing

Total RNA was extracted from 13 positive samples as described above. These samples represent 12 large dairy farms from different regions in Israel (Table 1). RNA extracted from these samples was used to amplify the S1 region of the spike glycoprotein and the full HE gene, using the Quanta qScript XLT 1-Step RT-PCR Kit (Qunatabio, Beverly, MA, USA). For S1 amplification, we used a six primers set previously published by Hasoksuz et al., 2002 [14], to amplify nucleotides 1–2731 of S glycoprotein of BCoV. HE full gene amplification was carried out using previously published primers [22]. RT-PCR products were loaded on 1.5% agarose gel and corresponding bands were gel extracted and purified by a MEGAquick Spin Plus kit (iNTRON, Seoul, Korea). Purified PCR products were verified by Sanger sequencing (Hylabs, Rehovot, Israel). Sequenced S1 and HE of local isolates were submitted to NCBI; accession numbers of local sequences are shown for each strain in phylogenetic trees.

### 2.7. Molecular Analysis of Local BCoV Strains

Nucleotide sequences of the HE and S1 regions from local isolates and selected reference strains were aligned with the Geneious alignment tool using Geneious software (Biomatters, CA, USA). These alignments were used to create a phylogenetic tree using MEGA X software [23]. A phylogenetic tree was produced based on the neighbor-joining method and bootstrapping with 1000 replicates and a support threshold of 70%. The deduced amino acid sequences were then assembled and aligned using BCoV Mebus strain as a reference and amino acid differences of local isolates were identified by the protein alignment tool in Geneious software.

## 3. Results

### 3.1. Clinical Signs and Molecular Detection of BCoV

During 2017–2021, a total of 853 fecal samples and rectum swabs were collected from large dairy farms scattered in Israel. These samples were screened for the presence of BCoV RNA by molecular detection using RT-qPCR and 250 samples were found to bepositive (29.3% of total samples). Among these positive samples, 87 showed severe symptoms of WD (around 10% of total samples). Following RT-qPCR, we selected 13 positive samples with Ct values ranging from 14.75 to 25.3, and further analyzed them (Table 1). These 13 samples represent 13 WD-diseased animals and were obtained from 12 large dairy farms in Israel (Table 1). Most of these animals were in the first and second lactation (Table 1) and showed distinct clinical signs of WD including fever, anorexia followed by hemorrhagic diarrhea, dehydration and subsequently, a significant drop in milk production (Table 1). Although WD is characterized by low mortality, its high morbidity rates usually lead to a substantial reduction in animal health and significant economic losses [24]. Most of the WD cases diagnosed in local farms were sporadic cases in adult cows (Table 1). However, two farms were affected the most, and BCoV outbreaks were observed in these farms (Table 1, farm #1 and #8). In farm #1, 57/700 (8%) adult dairy cows were diagnosed with WD clinical signs and in farm #8, 150/300 (50%) were affected with WD and eventually two animals died in this farm (Table 1). Overall, a 30% drop in milk production was reported in diseased cows in both farms.

### 3.2. BCoV Isolation and Replication

In order to explore and characterize local BCoV isolates, we used the HRT-18G cell line to culture BCoV virus from the feces of WD-positive samples (Figure 1). To this end, HRT-18G cells were cultured with isolate ISR2182. Three days post-infection, a cytopathic effect (CPE) was visible and characterized by the appearance of several dense clusters of enlarged and rounded cells (Figure 1, white arrows). We further verified BCoV replication in these cells by infecting HRT-18G with ISR2182 and performed RT-qPCR for BCoV RNA detection at different time points (0, 7, 24 and 44 h) (Figure 1B). Viral replication results indicate efficient viral replication, emphasized by the decrease in Ct values over the course of the experiment, shifting from Ct 29.45 at T0 to Ct 14.5 in T44 (Figure 1B). To support our RT-qPCR data, we performed immunofluorescence, targeting BCoV antigen replicating in HRT-18G cells (Figure 1C). To this end, we infected HRT-18G with isolate ISR2182 grown for 48 h and then analyzed viral replication and infectivity by immunofluorescence (Figure 1C). Our results show efficient replication and spreading of isolate ISR2182 compared to uninfected cells (Figure 1C).

### 3.3. Molecular Analysis

In order to characterize local BCoV isolates, we focused on molecular analysis of BCoV main attachment proteins, S and HE. Representative local BCoV-positive isolates were selected (Table 1). The S1 region of the Spike protein and full HE gene of these isolates were amplified and sequenced. Phylogenetic analysis of the S1 region nucleotides showed that local isolates cluster together with proximity to other strains from the Middle East and Europe, such as Egypt, Turkey and France (Figure 2A, KM386670, MK989630, KT318117 and MG757143, respectively). Importantly, the Mebus reference strain is clustered in a different branch; this result supports genetic distance analysis showing ~97% identity between Mebus and local strains. Moreover, a phylogenetic tree based on S1 nucleotides showed that local strains belong to several different variants (Figure 2A). For instance, isolates ISR2182 and ISR2348 clustered together as a distinct variant, while ISR1255 and ISR2382 belong to a different variant (Figure 2A). We next performed deduced amino acid sequence alignments of the S1 region of local isolates with the Mebus reference strain (Figure 2B). These alignments resulted in 56 amino acid sites that differ from Mebus (Figure 2B). Most of these amino acid changes were clustered in two domains, residues 33–257 and 458–571, Spike NTD and Spike RBD sites, respectively (Figure 2B). Interestingly, several amino acid changes found were common to all local isolates (e.g., positions 151–179, Figure 2B), while some strains showed polymorphism in amino acid sites that differed from other local strains (Figure 2B). For example, on farm #1, strains ISR2408 and ISR2391 are different from all the others in four distinct sites (242, 510, 769 and 776, Figure 2B). These data support nucleotides analysis (Figure 2A) and suggest that local strains belong to several genetic variants (Figure 2B). Next, we performed phylogenetic analysis of the HE gene (Figure 3). To this end, we sequenced the complete HE gene of local strains. Amplified sequences were used in alignment with selected HE sequences of reference strains (Figure 3A). Nucleotide analysis of the full HE gene revealed that local isolates cluster in a different branch from the reference strain Mebus, which is similar to and supports the S1 analysis (Figure 3A). Local isolates are closely related to HE of European strains (EF445643 and KX982264, Figure 3). Amino acid analysis of local strains compare to Mebus revealed overall 22 sites of amino acid substitutions, and some of them are similar in several strains (Figure 3B). Additionally, consistent with S1 analysis, HE amino acid analysis supports the existence of several genetic variants (Figure 3B). For instance, strains from farm #1 showed three amino acid distinct sites that differed from the reference strain Mebus and the other local strains (amino acid 181, 367 and 403).

## 4. Discussion

BCoV is the causative agent of WD in adult cattle. BCoV infection is usually identified by the appearance of hemorrhagic diarrhea and decrease in milk production of diseased cattle [6,9], which overall leads to significant economic losses. During 2017–2021, we detected several cases of WD in dairy farms in Israel. Some of the cases represent sporadic infections while others were part of BCoV outbreaks (Table 1). Although BCoV infections occurred in local farms, little is known about local BCoV enteric strains. Therefore, we aimed to detect, isolate and characterize local BCoV strains causing WD in Israel. To this end, we collected feces from 12 large dairy farms from different geographical areas in Israel; all samples showed distinct WD clinical signs and were found to be positive for BCoV by RT-qPCR (Table 1). Initial isolation of selected local strains on HRT-18G resulted in efficient replication, spreading of BCoV and finally induced significant changes in the monolayer, characterized by the presence of enlarged and dense cell clusters, which is in agreement with previous reports describing CPE induced by BCoV (Figure 1, [25]).

In Israel, the current strategy in preventing BCoV infection includes a recommendation to vaccinate adult cattle with commercial vaccines, which contain the enteric reference strain Mebus. However, several reports showed that current BCoV strains are genetically distant from the reference strain Mebus [14,22]. Additionally, reoccurrence of BCoV infections in vaccinated cattle in local farms was reported (Table 1). Therefore, we aimed to initially identify local BCoV variants and determine their genetic distance from selected reference strains (Figure 2 and Figure 3). To this end, we performed molecular analysis of the S1 hypervariable region and HE gene of local BCoV strains. Phylogenetic analysis of the S1 domain and HE gene revealed that local BCoV variants are distinct from the reference strain Mebus and clustered together with other BCoV circulating in the Middle East and Europe (Figure 2 and Figure 3). In addition, we found that local isolates belong to different variants regardless of geographical distributions, suggesting the movement of animals between farms in Israel (ISR1255 and ISR2382, Figure 2A). Overall, this data is consistent with several reports describing the genetic distance between Mebus and current BCoV strains [14,26]. Moreover, when compared to the Mebus strain, amino acid analysis of S1 from local isolates showed significant changes in numerous positions in Spike NTD and Spike RBD domains (Figure 2B). Some of the amino acid substitution positions found in local isolates were already reported in novel variants [14]. Interestingly, we found that in our analysis 26/56 sites were consistent with substitutions already published in variants that differ from Mebus. The additional variations in S1 were also clustered in the same major regions (amino acid 33–257 and 458–571). Overall, this suggests that these regions are subjected to selective pressure and might affect viral–host interactions. Yet, the effect of amino acid variations, found in local strains, on viral virulence and host response, and in particular, their effect on the protective immunity of adult cattle, needs to be tested by future studies. Although HE of BCoV is considered as a secondary receptor, its important role in the identification of novel emergence strains was recently published [27]. HE amino acid sequence analysis of local strains compared to Mebus revealed 22 amino acid substitutions (Figure 3B). None of these changes were common to all local strains; still, variations in HE amino acids of local strains support the results from S1 amino acid and nucleotides analysis, showing that several genetic variants are circulating in Israeli dairy farms.

## Figures and Tables

**Figure 1 viruses-13-01070-f001:**
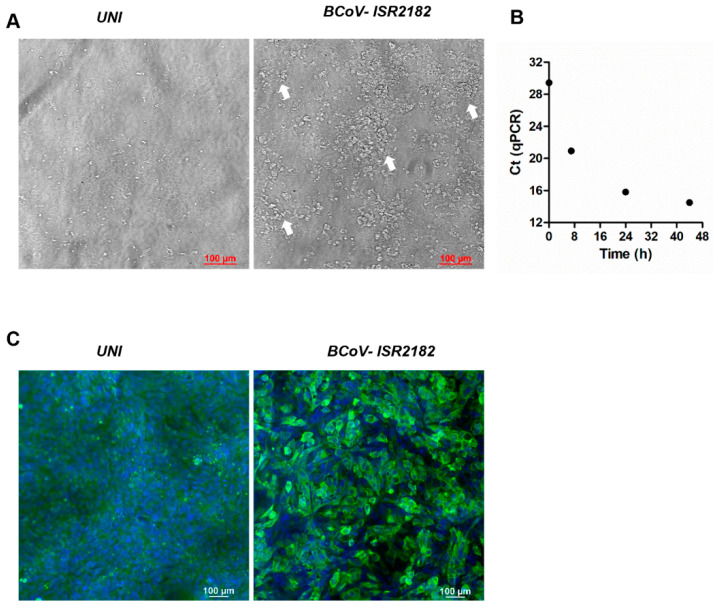
Cytopathic effect and replication of Bovine coronavirus (BCoV) isolate ISR2182. (**A**) Uninfected HRT-18G cells show intact monolayer, while cells infected with local isolate ISR2182 (1:10) for 72 h show significant changes in monolayer (white arrows). (**B**) Replication of BCoV ISR2182 in HRT-18G, cells were infected with isolate ISR2182 and collected at different time points. BCoV replication was measured by RT-qPCR. (**C**) Immunofluorescence images showing replication of BCoV isolate ISR2182 in HRT-18G cells as detected by monoclonal antibody-targeting coronavirus nucleocapsid protein, with DAPI for nucleus staining.

**Figure 2 viruses-13-01070-f002:**
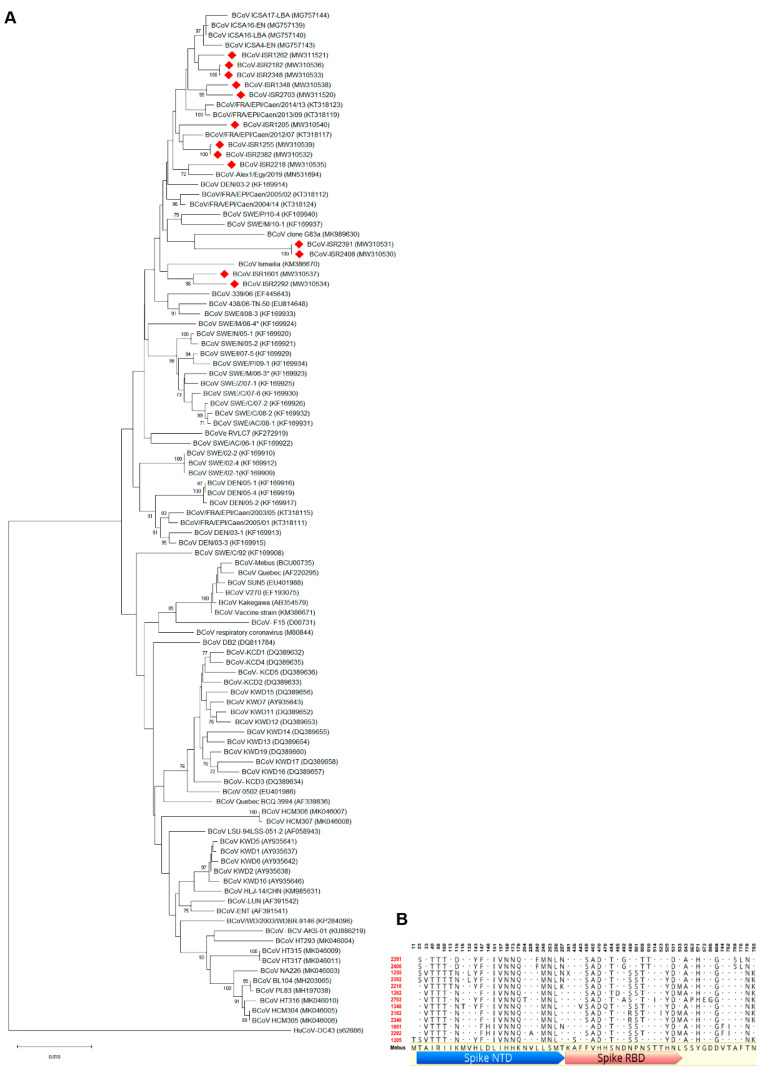
Phylogenetic tree of the partial Spike gene of Bovine coronavirus (BCoV) from local isolates. (**A**) Phylogenetic tree of S1 region of BCoV spike glycoprotein. BCoV genomic RNA was isolated from feces of WD diseased cattle and used for molecular characterization of the S1 region. Local isolates (labeled in red) were aligned with selected reference strains. Phylogenetic tree generated based on S1 nucleotide sequences (nucleotides 1–2731) was generated via the neighbor-joining method with bootstrap analysis (1000 replicates, >70%). The scale bar shows the number of substitutions per site. We used HCoV OC43 as an outgroup strain. Spike partial sequences were retrieved from GenBank and embedded in the figure for each strain. (**B**) Deduced amino acid sequences of S1 from local isolates aligned to reference strain Mebus. Local isolate sequences were aligned using Geneious protein alignment tool to show only disagreements residues compared to the reference sequence of Mebus strain.

**Figure 3 viruses-13-01070-f003:**
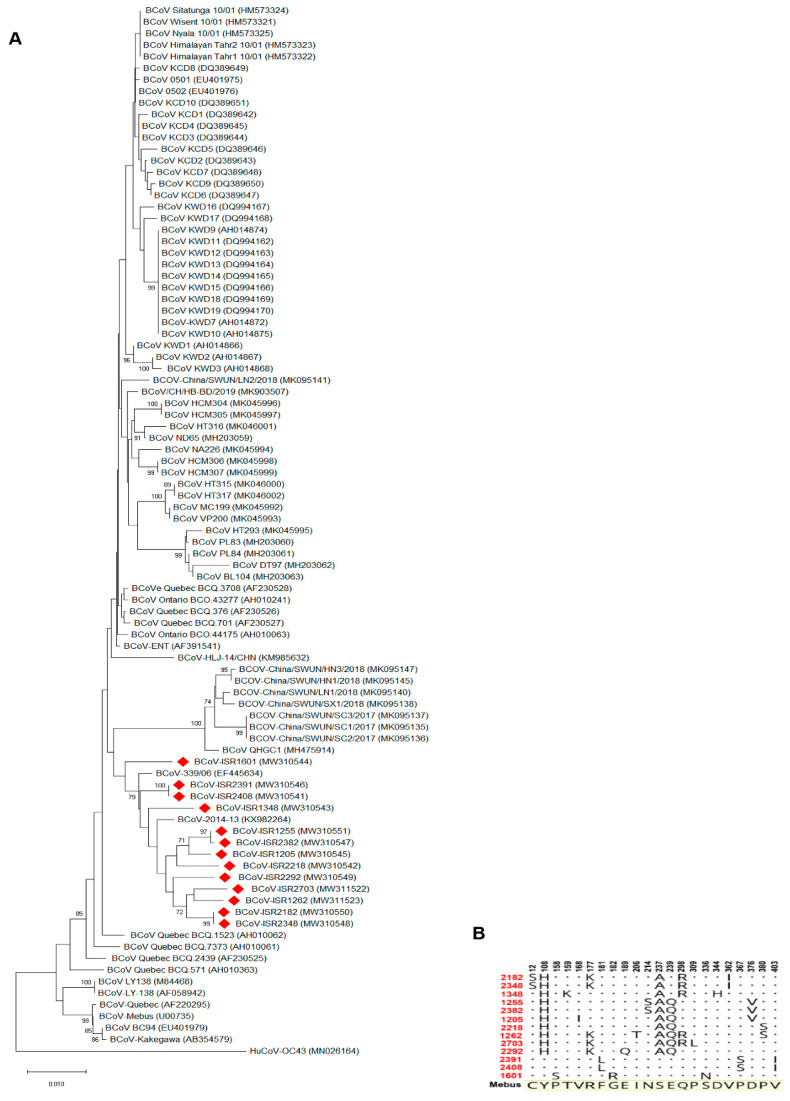
Phylogenetic tree of the Hemagglutinin Esterase (HE) gene of Bovine coronavirus (BCoV). Local isolates. (**A**) Local strains’ (labeled in red) alignment with selected reference BCoV HE sequences. Phylogenetic tree was generated based on full HE gene sequences. Phylogenetic trees were generated via the neighbor-joining method with bootstrap analysis (1000 replicates, >70%). The scale bar shows the number of substitutions per site. We used HCoV OC43 as an outgroup strain. HE sequences retrieved from GenBank and embedded in the figure for each strain. (**B**) Deduced amino acid sequences of local isolates aligned to reference strain Mebus. Amino acid alignment showing only disagreements residues compared to reference strain Mebus.

**Table 1 viruses-13-01070-t001:** Local Bovine Coronavirus isolated from winter dysentery-diseased cattle.

Farm (#)	Isolate *	Clinical Signs	Age	Year	C_t_ (RT-qPCR)	Prevalence ^€^
1	ISR2408	Hemorrhagic diarrhea, reduction in milk	Primipara, second and third lactation	2017	20.11	57/700
	ISR2391	Hemorrhagic diarrhea, reduction in milk	Primipara, second and third lactation	2017	21.92	
2	ISR1601	Hemorrhagic diarrhea	6 and 7 years old	2018	23.5	1/800
3	ISR1205	Hemorrhagic diarrhea, reduction in milk	Primipara, second lactation	2018	19.74	6/270
4	ISR2382	Hemorrhagic diarrhea, reduction in milk	Third lactation	2018	20.64	1/70
5	ISR1255	Hemorrhagic diarrhea	6 years old	2019	25.3	3/190
6	ISR1348	Hemorrhagic diarrhea, reduction in milk	In lactation	2019	14.75	6/370
7	ISR2218	Hemorrhagic diarrhea, reduction in milk	Primipara, second, third and fourth lactation	2020	23.71	15/320
8	ISR2182	Hemorrhagic diarrhea, reduction in milk, death	Primipara, second lactation	2020	16.91	150/300
9	ISR2292	Hemorrhagic diarrhea, reduction in milk, anorexia	Primipara, second, third and fourth lactation	2020	17.02	7/680
10	ISR2348	Hemorrhagic diarrhea, reduction in milk	In lactation	2020	20.44	1/350
11	ISR2703	Hemorrhagic diarrhea, reduction in milk	Primipara, second lactation	2020	23.23	3/350
12	ISR1262	Hemorrhagic diarrhea	Primipara, second, third and fourth lactation	2021	21.67	10/400

* Each isolate obtained from a single animal; ^€^ Animals with WD clinical signs from total numbers of cattle in each farm.

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
