# Peer review of "Characterization of Winter Dysentery Bovine Coronavirus Isolated from Cattle in Israel"

_viruses, 2021, doi:10.3390/v13061070_

Round 1

Reviewer 1 Report

#viruses-1234694

Comments to the Authors 

Authors collected fecal and rectal swab samples form dairy cattle showing Winter Dysentery (WD) clinical signs in Israel. They successfully isolated BCoV strains from 12 large dairy farms and conducted sequence analysis of the HE and partial S1 gene. From the phylogenetic analysis of S1 domain and HE gene, those isolated BCoV strains are distinct from the reference strain Mebus and clustered together with other strains circulated in the Middle East and Europe. These findings are very important and helpful for the control strategy using commercial vaccines. While the findings presented in this study should be of interest to Viruses readership, it requires moderate revision before ready for publication.

Moderate comments;

  1. Line 48-50: How many animals did show WD in each farm? You should add the information regarding the number of animals showing WD and its prevalence for each farm as Table.
  2. Line 115-117: How many samples/animals did you collect from each farm?
  3. Line 117-119: How many samples did you screen and finally detect 13 positive samples/animals?
  4. Line 126-128: You should describe the prevalence in not only farm #1 and 8, but also the other 10 farms (#2, 3, 4, 5, 6, 7, 9, 10, 11 and 12).
  5. Table 1: Were ISR24008 and ISR2391 isolated from one animal or different two animal?
  6. Discussion: From the result of Figure 3 and 4, ISR2382 (Farm #4) and ISR1255 (Farm #5) are almost same strain. ISR2182 (Farm #8) and ISR2348 (Farm #10) are as well. Is there any epidemiological relationship between Farm #4 and #5, and farm #8 and #10? For instance, cattle moved between those farms by selling. You should consider those results in Discussion.

Author Response

Reviewer #1:

We would like to thank reviewer #1 for his comments!

Please see below our point-by-point response:

Moderate comments;

  1. Line 48-50: How many animals did show WD in each farm? You should add the information regarding the number of animals showing WD and its prevalence for each farm as Table.

A: We thank reviewer # 1 for this comment. We add the total number of samples collected during these years. Please see revised line 48. We add the total numbers of BCoV positive samples and WD diseased animals in the results section line: 117-128. We believe the revised paragraph to be clearer to the readers.

  1. Line 115-117: How many samples/animals did you collect from each farm?

A: Please see our answer to comment 1.

  1. Line 117-119: How many samples did you screen and finally detect 13 positive samples/animals?

A: Please see our answer to comment 1.

  1. Line 126-128: You should describe the prevalence in not only farm #1 and 8, but also the other 10 farms (#2, 3, 4, 5, 6, 7, 9, 10, 11 and 12).

A: We thank reviewer #1 for this comment. We only mention farms #1 and #8 in our results part, due to the fact both suffered the most from WD outbreaks while others farms showed only sporadic cases. In our revised manuscript, we included these details in Table 1. Please see revised Table 1.

  1. Table 1: Were ISR24008 and ISR2391 isolated from one animal or different two animal?

A: These strains were isolated from two animals from the same farm- we add a comment to Table 1 to make it clear.

  1. Discussion: From the result of Figure 3 and 4, ISR2382 (Farm #4) and ISR1255 (Farm #5) are almost same strain. ISR2182 (Farm #8) and ISR2348 (Farm #10) are as well. Is there any epidemiological relationship between Farm #4 and #5, and farm #8 and #10? For instance, cattle moved between those farms by selling. You should consider those results in Discussion.

A: We thank reviewer #1 for this comment. We discussed these results in our revised manuscript. Line: 179-181 and line: 245-247.

Reviewer 2 Report

The manuscript presents molecular typing of bovine coronavirus isolated from clinical cases of winter dysentery in Israel. It is suggested to simplify the title of the manuscript by removing words "circulated" and "diseased" so the new title would be: Molecular characterization of bovine coronvirus isolated from winter dysentery cattle in Israel. Based on the title of this manuscript the study should encompass only genetic studies while authors included virus isolation, replication dynamics and IF detection. If this data should remain it is suggested to change the title of the manuscript. In Materials and Methods the authors did not provide the number of samples tested in total.  In reviewer's opinion figure 1 can be removed from the manuscript without negative impact on the content. The main shortcoming of this paper is low number of reference sequences used in phylogenetic studies. When partial S gene region was analyzed only 33 reference sequences from Genbank were included. It is even worse in case of HE gene coding region where only 9 sequences from GenBank were included. Additionally, only 5 of them were used also in analysis of S1 region. This low number of reference sequences can significantly influence the results of clustering. In other studies which are not cited in this manuscript (published in years 2017-2020 from Thailand, Vietnam, France and Japan) 50 reference sequences were used in phylogenetic analysis from 2017 but later studies included 95, 100 and 150 sequences from the Genbank. This way one can cluster isolates as it was done in other studies finding some geographic or disease related relationships. In this manuscript Israeli isolates cluster together with Egyptian and Turkish strains which is rather expected but the presence of French isolates is rather surprising. Using more reference strains from other regions would improve the quality of results obtained and their reliability. English grammar requires improvement and the support from the native speaker would be beneficial. The authors use singular and plural forms in the wrong way: line 13 (local strains belongs), line 25 (...lead...),  line 33 (...facilitate...), line 107 (...were...) and many other. Also the words are used in a wrong way: line 9 (...causing...), line 15 (...differ...), line 72 (full name for CPE is missing).

Author Response

Reviewer #2:

We would like to thank reviewer #2 for his helpful comments.

 Please see below our point-by-point response:

The manuscript presents molecular typing of bovine coronavirus isolated from clinical cases of winter dysentery in Israel.

  1. It is suggested to simplify the title of the manuscript by removing words "circulated" and "diseased" so the new title would be: Molecular characterization of bovine coronvirus isolated from winter dysentery cattle in Israel. Based on the title of this manuscript the study should encompass only genetic studies while authors included virus isolation, replication dynamics and IF detection. If this data should remain it is suggested to change the title of the manuscript.

A: We thank reviewer #2 for this comment. We changed the title. Please see the new title in our revised manuscript.

  1. In Materials and Methods the authors did not provide the number of samples tested in total. 

A: We included details regarding the numbers of animals and samples. Please see Line: 48 and Line: 117-128 in our revised manuscript.

  1. In reviewer's opinion figure 1 can be removed from the manuscript without negative impact on the content.

A: We agree with reviewer #2 and removed Figure 1 – please see our revised manuscript.

  1. The main shortcoming of this paper is low number of reference sequences used in phylogenetic studies. When partial S gene region was analyzed only 33 reference sequences from Genbank were included. It is even worse in case of HE gene coding region where only 9 sequences from GenBank were included. Additionally, only 5 of them were used also in analysis of S1 region. This low number of reference sequences can significantly influence the results of clustering. In other studies which are not cited in this manuscript (published in years 2017-2020 from Thailand, Vietnam, France and Japan) 50 reference sequences were used in phylogenetic analysis from 2017 but later studies included 95, 100 and 150 sequences from the Genbank. This way one can cluster isolates as it was done in other studies finding some geographic or disease related relationships. In this manuscript Israeli isolates cluster together with Egyptian and Turkish strains which is rather expected but the presence of French isolates is rather surprising. Using more reference strains from other regions would improve the quality of results obtained and their reliability.

A: We thank reviewer #2 for these comments. We initially focused on WD samples from GeneBank and therefore we used low numbers of reference sequences. Based on reviewer #2 comment we reanalyzed local isolates S1 and HE gene and used 103 sequences for S1 and 91 sequences for HE alignments analysis. We updated the figures with the new analysis. Please see revised Figures 2 and 3 in our revised manuscript. In addition, we inserted the references suggested by reviewer #2.

  1. English grammar requires improvement and the support from the native speaker would be beneficial. The authors use singular and plural forms in the wrong way: line 13 (local strains belongs), line 25 (...lead...),  line 33 (...facilitate...), line 107 (...were...) and many other. Also the words are used in a wrong way: line 9 (...causing...), line 15 (...differ...), line 72 (full name for CPE is missing).

A: We thank reviewer #2 for this comment. We checked the manuscript for language again, fixed issues, and uploaded a revised manuscript.